# Testing the Performance of an Innovative Markerless Technique for Quantitative and Qualitative Gait Analysis

**DOI:** 10.3390/s20226654

**Published:** 2020-11-20

**Authors:** Laura Simoni, Alessandra Scarton, Filippo Gerli, Claudio Macchi, Federico Gori, Guido Pasquini, Silvia Pogliaghi

**Affiliations:** 1Department of Neurosciences, Biomedicine and Movement Sciences, University of Verona, 37129 Verona, Italy; laura.simoni@univr.it; 2IRCCS Fondazione Don Carlo Gnocchi ONLUS, 50143 Florence, Italy; fgerli@dongnocchi.it (F.G.); claudio.macchi@unifi.it (C.M.); gpasquini@dongnocchi.it (G.P.); 3Microgate SRL, 39100 Bolzano, Italy; alessandra.scarton@microgate.it (A.S.); federico.gori@microgate.it (F.G.)

**Keywords:** gait analysis, treadmill walking, markerless systems, harmony, Fast Fourier Transform

## Abstract

Gait abnormalities such as high stride and step frequency/cadence (SF—stride/second, CAD—step/second), stride variability (SV) and low harmony may increase the risk of injuries and be a sentinel of medical conditions. This research aims to present a new markerless video-based technology for quantitative and qualitative gait analysis. 86 healthy individuals (mead age 32 years) performed a 90 s test on treadmill at self-selected walking speed. We measured SF and CAD by a photoelectric sensors system; then, we calculated average ± standard deviation (SD) and within-subject coefficient of variation (CV) of SF as an index of SV. We also recorded a 60 fps video of the patient. With a custom-designed web-based video analysis software, we performed a spectral analysis of the brightness over time for each pixel of the image, that reinstituted the frequency contents of the videos. The two main frequency contents (F1 and F2) from this analysis should reflect the forcing/dominant variables, i.e., SF and CAD. Then, a harmony index (HI) was calculated, that should reflect the proportion of the pixels of the image that move consistently with F1 or its supraharmonics. The higher the HI value, the less variable the gait. The correspondence SF-F1 and CAD-F2 was evaluated with both paired t-Test and correlation and the relationship between SV and HI with correlation. SF and CAD were not significantly different from and highly correlated with F1 (0.893 ± 0.080 Hz vs. 0.895 ± 0.084 Hz, *p* < 0.001, r^2^ = 0.99) and F2 (1.787 ± 0.163 Hz vs. 1.791 ± 0.165 Hz, *p* < 0.001, r^2^ = 0.97). The SV was 1.84% ± 0.66% and it was significantly and moderately correlated with HI (0.082 ± 0.028, *p* < 0.001, r^2^ = 0.13). The innovative video-based technique of global, markerless gait analysis proposed in our study accurately identifies the main frequency contents and the variability of gait in healthy individuals, thus providing a time-efficient, low-cost means to quantitatively and qualitatively study human locomotion.

## 1. Introduction

Human mobility is a fundamental requirement for a satisfactory quality of life [1]. In particular, the ability of walking is certainly one of the key requirements for many activities of daily life. Through quantitative gait analysis, precious information can be acquired to assess subjects’ functional capabilities and to prescribe individualized therapeutic interventions [2,3,4].

Among the many relevant measurements, spatiotemporal parameters are widely used both in research and clinical context. These quantitatively describe the main events of the gait cycle (e.g., stance and swing phase) that reflect the ability of the subject to fulfil the general requirements of walking (e.g., balance, symmetry, coordination) [5]. Abnormalities of spatiotemporal parameters such as an excessive stance duration, stride frequency or cadence (SF—stride/second, CAD—step/second), or stride variability (SV) are fundamental to diagnosing pathologic gait and to monitor functional outcome after rehabilitation treatments [2,5,6,7]. For example, spatiotemporal parameters are largely used to assess the risk of fall in older adults and to evaluate the progression of gait impairments in various degenerative movement disorders such as myopathy, multiple sclerosis and Parkinson’s diseases [8,9].

Human gait can be further characterized in terms of its quality or harmony, an essential feature warranting efficient, smooth and safe movements [10]. Harmony (also called harmonicity) in human locomotion has been related to the functionality and health of the overall neuromuscular system [8,11]. For example, a low harmony of the center of mass acceleration during walking is associated with a higher risk of falling in older adults and a lower walking economy; moreover, the coordination of the principal body segments and joints in running gait has been shown to demarcate healthy from injured runners [11,12,13,14]. Despite the use of the term harmony in various fields of gait analysis, reference methods and standard indexes for the measurement of gait quality are still vaguely explored [10,15,16,17,18,19]. Usually, gait harmony is evaluated by considering the variability of spatiotemporal gait measures, their asymmetries between left and right side and with the step-by-step rhythmicity of acceleration patterns of the center of mass (i.e., the Harmonic Ratio) [16,17,18,20,21,22].

Modern quantitative and qualitative gait analysis is usually performed in well-instrumented, specialized laboratories by using an optical motion capture system, which is considered the gold standard method [3,4,6]. During a gait analysis trial, kinematics and kinetics of the main body segments and joints are computed respectively by means of 200 Hz cameras and force platforms and then evaluated—either by comparison with the reference values of healthy people or by identifying and monitoring of specific biomechanical markers of dysfunction [2,4]. Major drawbacks of the optical motion capture system are the high costs and the time investment and complexity of experimental setup, data acquisition and processing [7]. These are some of the reasons why gait analysis is widely used for research but is not the standard of care in clinical medicine worldwide.

Low-cost, time-efficient and non-invasive alternatives to traditional technologies (e.g., photoelectric sensors, inertial sensors, video-based markerless motion analysis methods) have been used for the large-scale diagnosis/longitudinal quantitative monitoring of gait abnormalities [23]. Furthermore, the study of the biomechanics of separate body segments, that represent the entire body, and/or the evaluation of the interaction, coordination and symmetry of multiple body segments have been applied for the qualitative analysis of gait through these alternative techniques [9,16,23,24,25,26]. Moreover, based on the notion that the human body is a complex system with multiple segments that move simultaneously during gait, recent studies propose global approaches that may offer a time-efficient, synthetic view of the harmony and quality of gait [15,27]. For example, the variability and left to right symmetry of the body silhouette has been used to quantify posture stability and symmetry of walking [15,28,29,30]. Moreover, the variability of the displacement on the sagittal plane of 22 markers placed on the whole-body has been used [15]. However, the search for a simple, time- and cost-efficient method to analyze harmony of cyclic movements is still open [12,15].

In this context, we developed a web-based video analysis software which automatically elaborates videos recorded by a commercial 60 Hz camera, providing as a result the value of quantitative gait indexes (stride frequency and cadence) and an index of gait quality or harmony (harmony index—HI). The calculation of these indexes relies on the spectral analysis of the variation of brightness of the individual pixels that compose a subject’s image, during the action of walking on a treadmill. The above analysis relies on the assumption that in a video of a harmonic movement (i.e., walking gait), the greater part of the points of the image of the subject’s body move in unison and that the main “forcing” frequency of their motion can be identified based on variations of pixels’ brightness. We tested the hypothesis that this approach could provide an accurate, non-invasive, time- and cost-efficient and easy to use method for the quantitative and qualitative analysis of gait, by comparison with a validated photoelectric sensors gait analysis system.

## 2. Materials and Methods

In this study, we analyzed spatiotemporal gait parameters and the frequency content of gait and its harmony from 86 healthy individuals (66 males and 20 females). Patients had a mean(SD) age of 32(14) years, a weight of 69.7(13.0) kg, height of 173.7(8.1) cm and a body mass index (BMI) of 22.7(4.0). Data were collected at two medical and research centers (Pro Motus, Bozen and IRCCS Don Gnocchi, Firenze, Italy) from patients involved in various clinical and research activities. This study was approved by the Ethics Committee of the Don Gnocchi Foundation (number 13663_oss) and all methods were performed in accordance with relevant guidelines and regulations of the institution. Informed consent was obtained from all individual participants included in the study. Inclusion criteria were: age range between 18–65 years, the ability to ambulate independently and absence of medical and musculoskeletal conditions that might affect the normal gait pattern causing abnormal motor response fluctuation (e.g., Alzheimer’s, Parkinson’s, dementia, osteoarthritis). Furthermore, subjects on pharmacological treatments that could influence motor control were excluded. All subjects were preliminarily adapted in the use of a treadmill (i.e., 1 h of total practice) [31]. The test consisted in performing a 90sec walk on a flat motorized treadmill (MTC-Climb, Runner, Italy and Woodway, USA) at self-selected speed. This task was preceded by 5 min of familiarization: treadmill velocity was initially set at 0.5 km/h and modulated until a subjectively comfortable speed was reached and recorded as the self-selected speed [32]. Subjects walked at an average self-selected speed of 1.1(0.1) m·s^−1^, with a Stride Length (SL) of 123.8(13.3) cm and a normalized SL of 0.71(0.07). Normalized SL (nSL) was also measured as the ratio between SL and height.

Spatiotemporal gait parameters were analyzed through the use of a previously validated photoelectric sensors system (Optogait, Microgate) [25,33] and a custom-designed web-based video analysis software (Graal, Microgate, Bolzano), used also for research purposes [34,35]. All spatiotemporal gait parameters were recorded during the 90” walking test and the analysis that follows was performed on the entire test.

Optogait is a system for optical detection made by a transmitting and a receiving 1-m bar placed on the sides of the treadmill tape, at the contact surface level, connected to a computer controlled by the researcher. Each bar contains 96 light-emitting diodes (LEDs) communicating on an infrared frequency with the same number of LEDs on the opposite bar. Once positioned on the treadmill, the system detects the interruptions of the communication between the bars (caused by the subject’s movement) and calculates the duration and position. During the execution of a gait, contact and stride times can be measured with an accuracy of 1 thousandth of a second and the position of the interrupted LEDs with a space resolution of 1.041 cm. Data were recorded continuously, at a frequency of 1000 Hz, through the dedicated software provided with the system which measures in real-time a series of crucial data for the movement analysis, such as step length, stride length, cadence, stride time and velocity. Software analysis settings for the identification of gait events were optimized based on pilot testing with concurrent video recording choosing a 1_1 filter setting. The parameters measured directly by the Optogait system were stride length (SL), nSL, SF, CAD and SV. SL, in cm, is defined as the distance between two consecutive initial foot contacts of the same foot (i.e., right foot – right foot). SF, defined as the number of gait cycles per second, and CAD, defined as the number of ground contact events per second [25,30], were identified for each patient, based on the duration of the stride and of the step phases of gait and the following formulas:(1)$$\mathrm{SF}\left[\mathrm{Hz}\right]=\frac{1}{\mathrm{Stride}\;\mathrm{duration}\;\left[s\right]}$$
(2)$$\mathrm{CAD}\left[\mathrm{Hz}\right]=\frac{1}{\mathrm{Step}\;\mathrm{duration}\;\left[s\right]}$$

Successively, the within-subject coefficient of variation of SF was calculated on the entire test duration, to measure SV as an index used for gait harmony [16,26,36,37].

Concurrent video recordings were registered at 60 Hz and with a resolution of 640 × 480 pixels with a Logitech Brio 4 K synchronized with the optoelectronic system. A camera Logitech Brio 4 K was synchronized with the Optogait system and it was used to register videos at 60 Hz of the 90” walking tests, with a resolution of 640 × 480 pixels. The webcam was placed behind the patient subjects at a distance of 2.5 m and with a height from the ground of 95 cm. Ambient were well-illuminated by standard artificial light with a top-down direction. The custom-designed web-based video analysis software (Graal, Microgate) was used to automatically perform a spectral analysis of the changes over time in the brightness for each pixel of the gait video. This analysis is intended to provide values of the frequency content of gait and of its harmony. The first step of the analysis is the subdivision of the video file into frames, in order to obtain a given number (*n*) of individual images in JPEG format. For each pixel of the image at a given time, the software computes the brightness by using the following brightness editing algorithm [38]:(3)$$B=\frac{\left(r+g+b\right)}{3}$$
where B is the brightness value, and r, g, b are the coordinates of the image according to the Red, Green and Blue (RGB) color model.

Then, the Fast Fourier Transform (FFT) algorithm is applied to the vector of brightness values over time for each pixel in order to obtain its magnitude M in the frequency domain:(4)$$M=\frac{\mathrm{abs}\left(\mathrm{fft}\left(B,n\right)\right)}{\frac{n}{2}}$$
where abs is the absolute value of the FFT of the brightness signal and n is the number of frames.

As a result, a power spectrum for each pixel is generated that displays the magnitude as a function of frequency and allows to identify the peaks corresponding to the dominant frequencies of variation over time. A simple peak detection is thus performed to extract the peaks with the highest power between 0–15 Hz. This upper frequency limit was chosen to allow the evaluation of different forms of locomotion, from walking (characterized by an average frequency content around 2 Hz [23]) to running (characterized by an average frequency content around 2.6–2.8 Hz for velocity of 13 Km/h [24]), while excluding information that is not relevant. Typically, up to 10 peaks or “interesting frequencies” for each pixel are identified, characterized in frequency and magnitude. The frequency peaks from all pixels of the image are then appended in a single list of interesting frequencies based on which a frequency distribution plot is generated (i.e., histograms counting the number of times a given frequency is observed versus the frequency itself). This representation provides a visual synthesis of the distribution of dominant frequencies describing the motion captured in the video. Histograms with few distinct and pronounced peaks would indicate a well-coordinated and smooth movement (i.e., harmonic) (Figure 1A). By contrast, a larger number of blurred peaks would indicate jittery or uncoordinated movement (such as walking in a clinical population) (Figure 1B).

We identified the first (F1) and second (F2) most important frequencies as those associated with the largest % of pixels.

The list of significant frequencies was then sorted by frequency and plotted as a function of line number. This allows to display a graphic where similar frequencies are grouped: in the case of frequencies appearing in a large number of pixels, clear “steps” appear in the graphic (Figure 2A, blue line), indicating that a large proportion of pixels of the image move consistently; on the contrary, when different frequencies appear in a small number of pixels, a smoother increasing profile results (Figure 2B, blue line). We speculated that the higher the proportion of the pixels that move in unison with the forcing frequencies of gait (which would reflect a harmonic pattern), the higher the “steppiness” of the “step plot” (Figure 2). To quantify this quality, we created a measure of harmony or harmony index (HI), calculated as the fitting error of a polynomial of degree 15 to the “step plot” data points:(5)$$\mathrm{HI}=\frac{\mathrm{sum}\left(\mathrm{abs}\left(f-y\right)\right)}{N}$$
where f is the polynomial function and y is the sorted list plot. N is the number of data points.

The higher the value of HI, the less variable is the motion under analysis.

The first step of the statistical analysis was to check normality of the data by visual inspection and by using the Shapiro–Wilk test. Data exhibited a normal distribution. Group average and ± SD (standard deviation) were calculated for all variables. Within subject average ± SD for SF and CAD were calculated on the entire 90” of the test and SV was obtained as within-subject coefficient of variation of SF. Averages of gait parameters acquired by the Optogait system were calculated for males and females. Measures of stride frequency, cadence and gait harmony obtained by the Optogait system were compared to those measured trough Graal video analysis. The correspondence between SF and F1 and CAD and F2 was evaluated by paired t-Test, Bland-Altman analysis followed by one-sample t-Test and Pearson product-moment correlation analysis. The association between SV and HI was tested with Pearson product-moment correlation analysis. Data were analyzed using SigmaPlot 11 (Systat Software Inc., San Jose, CA, USA).

## 3. Results

Table 1 reports mean (SD) values for spatiotemporal parameters measured with the Optogait system, for males and females.

Successively, the comparison between the Optogait system and Graal was carried out to assess the correspondence between SF and F1 and CAD and F2 and the association SV vs. HI. SF and CAD were not significantly different from F1 (0.893 ± 0.080 Hz vs. 0.895 ± 0.084 Hz, *p* < 0.001) and F2 (1.787 ± 0.163 Hz vs. 1.791 ± 0.165 Hz, *p* < 0.001). A non-significant bias and a very small imprecision was confirmed by the Bland-Altman test between F1 and SF (Figure 3B). A significant yet practically barely measurable bias was associated with the comparison between F2 and CAD, with an extremely small imprecision (Figure 3B). Moreover, SF and CAD were highly correlated respectively with F1 (0.893 ± 0.080 Hz vs. 0.895 ± 0.084 Hz, *p* < 0.001, r^2^ = 0.99) and F2 (1.787 ± 0.163 Hz vs. 1.791 ± 0.165 Hz, *p* < 0.001, r^2^ = 0.97) (Figure 3A). In addition, the SV was 1.84% ± 0.66% and it was significantly and moderately correlated with HI (0.082 ± 0.028, *p* < 0.001, r = 0.36, r^2^ = 0.13) (Figure 4).

## 4. Discussion

The purpose of this study was to evaluate, in healthy individuals, the performance of Graal, an innovative, custom-designed, web-based video analysis software, with the aim of identifying the main quantitative and qualitative characteristics of normal gait in a further simplified and comprehensive approach. The results show how Graal can accurately identify the main frequency contents and gait variability in healthy individuals by measuring F1, F2 and HI: F1 is not significantly different from SF and it is highly correlated with it, and the same is true for F2 vs. CAD. Finally, HI correlates with SV.

The values of SL, SF and CAD of our sample are consistent with those reported in previous studies that analyze the walking gait of healthy adults at similar self-selected speeds [23,39,40,41,42], thus confirming the results already present in literature. On the other hand, to the best of our knowledge, there are no other studies that provide treadmill walking gait values for healthy adults of nSL and SV.

Given the quite large number of patients in our sample (86) and their wide age range of 32(14) years, the results of this study could provide reference values for spatiotemporal parameters of walking gait in healthy adults of both sexes, in particular providing normal values for nSL and SV. SF, CAD, SL and SV are basic qualitative parameters of spatiotemporal gait analysis that give important information on gait stability both among elderly and pathological individuals [4,37].

To the best of our knowledge, our video-based system Graal is the first method able to estimate, with a good reliability and almost automatically, stride frequency and cadence either without markers or sensors on the patient’s body, or without calibration and post-processing manual editing of the video [28,29]. The above video-based markerless approach to gait analysis had been previously proposed by Stone and Skubic, Clark et al. and other research groups [29,43]. However, the above studies reported some imprecision in the detection of spatiotemporal parameters, due to the difficulty in the initial contact detection. The main reason for the higher accuracy of our method, compared to pre-existing video analysis systems, is the estimation of stride frequency and cadence as the two main frequencies of variation of pixel brightness over time (F1 and F2), without the need to identify the initial contact for each step from the video [30,43,44,45,46].

With regard to the spatial parameters stride and step length, our video-based system Graal does not provide a direct measure; however, these parameters can be easily calculated based on *F*1 (equivalent to stride frequency) and *F*2 (equivalent to cadence) as follows:(6)$$\mathrm{Stride}\;\mathrm{length}\left(m\right)=\mathrm{speed}\;\left(\frac{m}{s}\right)\;\times \;F1\;\left(\mathrm{Hz}\right)$$
(7)$$\mathrm{Step}\;\mathrm{length}\left(m\right)=\mathrm{speed}\;\left(\frac{m}{s}\right)\;\times \;F2\;\left(\mathrm{Hz}\right)$$

In our study, the significant, low-to-moderate correlation (r = −0.36) between SV and HI is compatible with the hypothesis that HI could represent a valid index of gait harmony. The value of SV in our study (1.84% ± 0.66%, range 0.91–3.12%) was consistent with literature norms in a similar population and is indicative of a fully physiological gait (i.e., <4%) [47,48]. It is plausible, yet it remains to be verified, that the inclusion of individuals with a wider range of gait variability and/or an abnormally variable gait would increase the strength of the association between SV and HI.

Traditionally, SV or other indices derived from Center of Mass (CoM) acceleration (i.e., harmonic ratio, smoothness) are used to describe gait harmony, even if the measure is indirect and obtained from a segmental approach [16,18,26,36,37]. Compared to SV or the other traditional harmony measures, HI has the important advantage of considering the contribution of all body segments and joints in the quality of gait. The use of an index of harmony derived from the description of the gait frequency content trough the Fast Fourier Transform is based on the intuition that normal gait is a nearly periodic signal and anomalies usually disturb such periodicity [23]. An attempt to quantitatively describe the global gait harmony trough an approach similar to that shown in our study, was made by Williams and Vicinanza in 2017 [23]. The method they propose consists in the evaluation of the main frequency content of gait, through spectral analysis of the movement of 22 markers placed bilaterally on both the lower and upper body. Through a quantitative description of how a certain spectrum of movement frequency was deviating from a perfectly harmonic one, an “index of inharmonicity” was proposed [13]. Even if Williams and Vicinanza’s approach is based on the same assumption as the one presented in our study, it has not been validated and has only been described as applied to a case study of a single young adult. Thus, HI is the only validated index that provides a quantitative measure of the overall harmony of gait.

Finally, the availability of a quantitative index, that provides a global picture of gait, could be particularly useful in the evaluation of the progression of specific neurodegenerative diseases (e.g., Parkinson’s disease, Multiple Sclerosis), characterized by an alteration in the planning of the global gait movement, rather than the degeneration of specific patterns [4,11,28]. Generally, the progression of these diseases is recorded mainly through qualitative analysis, based on anamnestic reports or clinical evaluation scales (e.g. Expanded Disability Status Scale—*EDSS*, Unified Parkinson’s Disease Rating Scale—*UPDRS*), which do not have the same sensitivity at all stages of the disease. Future studies could evaluate the performance of Graal in the evaluation and periodical monitoring of different forms of physiological and pathological gait.

In conclusion, Graal is an innovative, valid, cost- and time-efficient method that measures some but not all of the spatiotemporal parameters that can be used for gait analysis in the clinic (e.g., it cannot measure step width or stance phase). The indexes provided (i.e., stride frequency, cadence, step/stride length, index of harmony) allow first-level evaluation and monitoring of gait and may inform the indication for a second-level investigation. From a practical stand-point, time-efficiency is a key aspect: all it takes for analysis and report procedures is a 30 s video recording of treadmill walking, that can be uploaded and analyzed in about 10 min. At the moment, the main known limitation of this methodology is that it can only be used indoors. The method is based on the spectral analysis of the fluctuations of brightness over time of the pixels that compose the image of a person while walking. As such, the method needs stable, artificial light conditions. In the well-illuminated ambient room used for the current study, the performance of the method is independent from the absolute ambient light. However, the validity of this approach in a sub-optimally illuminated or else in an unusually bright place remain to be verified.

## Figures and Tables

**Figure 1 sensors-20-06654-f001:**
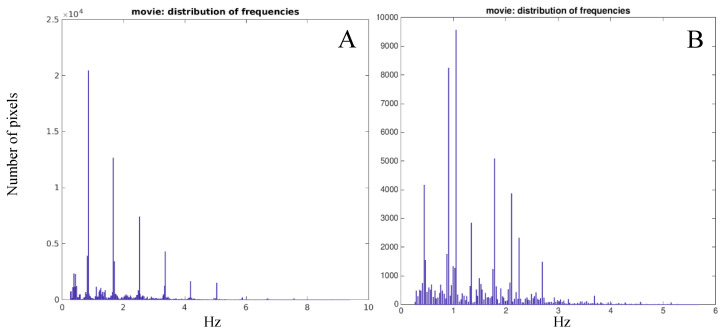
Panel (**A**) shows the histogram of a person walking with a well-coordinated motion pattern. Panel (**B**) shows the histogram of a man affected by Parkinson’s disease walking.

**Figure 2 sensors-20-06654-f002:**
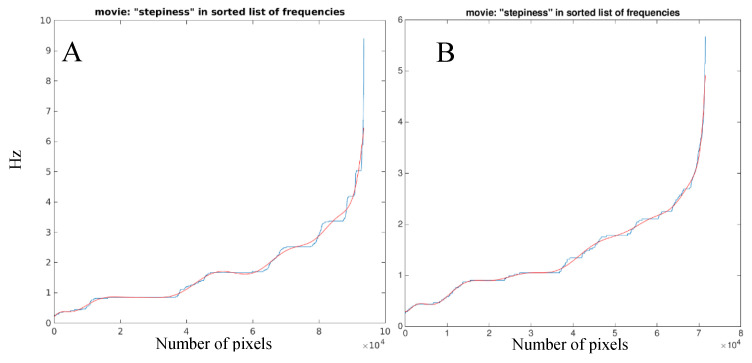
Figure (**A**) shows the “step plot” in blue and the polynomial fitting function in red of a man walking with a well-coordinated, “steppy”, motion pattern. Figure (**B**) shows the irregular, “smoother”, gait of a man affected by Parkinson’s disease. It is possible to observe less, well-demarcated frequency components in case A compared to many “blurred” components in case B.

**Figure 3 sensors-20-06654-f003:**
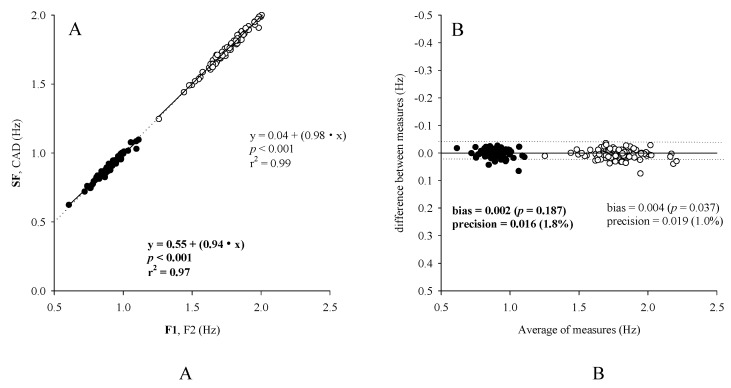
Scatterplots (**A**) and Bland-Altman plots (**B**) of stride frequency (SF) and Cadence (CAD) vs Frequency 1 (F1) and Frequency 2 (F2), measured by Optogait against Graal.

**Figure 4 sensors-20-06654-f004:**
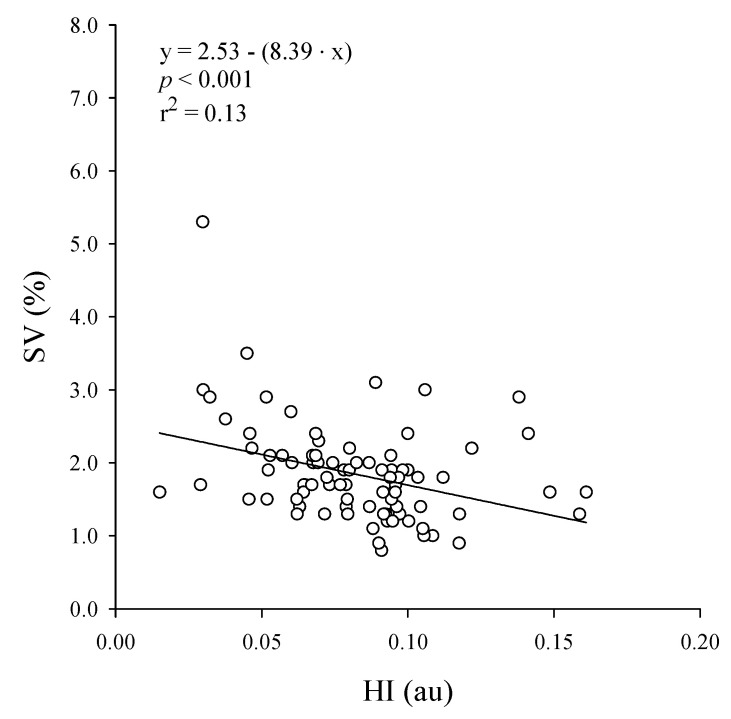
Correlation between Stride Variability (SV) and Harmony Index (HI).

**Table 1 sensors-20-06654-t001:** Values of spatiotemporal gait parameters measured with the Optogait system. Mean (standard deviation).

Patients	Males *n* = 66	Females *n* = 20	Total *n* = 86
Speed (km/h)	1.1 (0.1)	1.01 (0.1)	1.1 (0.1)
Stride length (cm)	126.8 (12.5)	114.9 (11.1)	123.8 (13.3)
Normalized stride length	0.72 (0.06)	0.69 (0.07)	0.71 (0.07)
Stride frequency (Hz)	0.89 (0.08)	0.89 (0.08)	0.89 (0.08)
Step frequency or Cadence (Hz)	1.79 (0.16)	1.79 (0.18)	1.79 (0.16)
Stride variability	1.78 (0.57)	2.04 (0.90)	1.84 (0.66)

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
