# Peer review of "Testing the Performance of an Innovative Markerless Technique for Quantitative and Qualitative Gait Analysis"

_sensors, 2020, doi:10.3390/s20226654_

Round 1

Reviewer 1 Report

Authors proposed and tested an innovative markerless techinque to perform quantitative and qualitative gait analysis. The letter is in line with the journal aim and it is well structered. The introduction provide useful information to understand the pros and cons of the markerless gait analysis. The methodology is well designed and described, as well the results are clearly reported. 

I believe that the manuscript is suitable for pubblication as letter. I have only few minor comments:

  1. I suggest to report some references related to the use of optoGAIT for research applications.
  2. I strongly suggest to use s rather than sec to indicate the measure unit of time according to the SI.
  3. Did author take into account the different size of the man and female group when performing statistical comparison? Please discuss
  4. Did author consider the goodness of the decimal point reported for the frequency? Is the methodology able to quantify the mHz?
  5. Please check the format of the results: mean +- SD unit of measure is not coherent with the SI. The solution is report the unit of measure for both mean and SD or include parenthesisi before mean and after SD.

Author Response

We wish to thank the Editor and the reviewers for the time and effort invested in reviewing our manuscript and for the constructive criticism.

We did our best to respond to the reviewer’s comments and to incorporate the suggestions that were offered.  Point-by-point answer to the reviewer are provided below. The reviewer’s comments are reported in bold, changes made to the manuscript are highlighted in yellow in the manuscript text.

Authors proposed and tested an innovative markerless techinque to perform quantitative and qualitative gait analysis. The letter is in line with the journal aim and it is well structered. The introduction provide useful information to understand the pros and cons of the markerless gait analysis. The methodology is well designed and described, as well the results are clearly reported. 

The reviewer has correctly summarised the aims and main findings of or study. In fact, we decided to incorporate the suggested term “markerless gait analysis” throughout the text, starting right from the title; we think that this smart suggestion will facilitate the immediate comprehension of the proposed method, possibly overcoming part of the confusion lamented by reviewer #3.

I believe that the manuscript is suitable for pubblication as letter. I have only few minor comments:

  1. I suggest to report some references related to the use of optoGAIT for research applications.

We thank the reviewer for the suggestion. At line 131 we specified that Optogait system is used also for research applications and we added some references. We also added more detail in the methods section, as per reviewer #3 request.

  1. I strongly suggest to use s rather than sec to indicate the measure unit of time according to the SI.

Thanks for the correct remark. We replaced “s” with “sec” according to the SI.

  1. Did author take into account the different size of the man and female group when performing statistical comparison? Please discuss

The reviewer correctly points out that differences in either variance and/or sample size needs to be accounted for when different groups are compared. We actually did so by applying a t-test for unequal variance samples. Having said this, also based on the suggestion of reviewer # 3, we have realised that we may have excessively emphasised the male/female comparison. This is in fact not the focus of the study. Therefore, while we presented the spatiotemporal gait parameters separately for males and females to allow the comparison with the literature, we have decided to remove the direct statistical comparison between sexes.

  1. Did author consider the goodness of the decimal point reported for the frequency? Is the methodology able to quantify the mHz?

The reviewer correctly enquires on the criterion used for the choice of how many decimal places to report. This typically depends on the resolution of the measurements and/or practical considerations. In this case the new markerless methodology is able to quantify not only mHz but seven decimal points after the comma. In addition, the Optogait photo-sensor technology has a mHz resolution. However, we choose to report only two decimal places because this is more typically the format chosen by most publications on gait analysis in clinical medicine. 

  1. Please check the format of the results: mean +- SD unit of measure is not coherent with the SI. The solution is report the unit of measure for both mean and SD or include parenthesisi before mean and after SD.

Thanks for the suggestion. We replaced +- SD, including parenthesis before mean and after SD.

Reviewer 2 Report

Dear Authors,

Thank you for submitting your ms. It is advised that the various annotations be addressed. 

A thorough grammar and spell check would improve the quality of the paper.

Author Response

We thank the reviewer for the useful feedback and for the grammar and spelling annotations. We have made all the changes as suggested in the PDF provided by the reviewer.

Reviewer 3 Report

The authors present a video-based software that can calculate some parameters, called F1 and F2, which are correlated to step and stride frequency, stride variability and gait harmony. I have a lot of issues in understanding their methodology and how F1 and F2 are clinically relevant for clinicians. I don't really understand the need for their software and the validation of their software output is questionable.

Below are the comments

Abstract:
You should define stride frequency (SF) and step frequency (SPF) in the abstract.
You forgot to define CV

Introduction:
l 42-45: This one sentence is a bit long and complicated and does not make much sense to me. Can you try to break it down.
line 50: You should define stride frequency (SF) and step frequency (SPF)
line 59: Missing reference after your statement on harmony associated with risk of falling in older adults.
line 61: I would not say that the term harmonicity is common in gait analysis
l 66-68: I am really surprised to hear that stereophotogrammetry is a gold standard in gait analysis. I have been working in that field for some time and never ever heard of it for gait analysis. Moreover the authors cite 3 papers to ascertain their point but non of the 3 papers refer to stereophotogrammetry systems.
l 66 - 74: I believe you are confusing 2D and 3D gait analysis. With 2D gait analysis you can easily extract spatiotemporal parameters (you don't even need cameras just a pressure mat will be enough). However to describe kinematics and kinetics (3D gait analysis) you will need an optical motion capture system with spherical markers taped to specific body landmarks (or active markers which is less common). That will be your 3D gait analysis gold standard to compute kinematics. Then you can add force plates to compute kinetics. I hope it makes more sense than your paragraph.
l 69: What do you mean high frequency? Usually the cameras will record at 200Hz (which I believe is not that high frequency, you can lower it down to 100Hz if needed). The force plates on the other hand needs really high frequency to capture heel strike (usually 1000Hz).
l 73-74: Most major clinics have an optical motion capture system and this system was initially developed for clinics and then was implemented in research facilities and for the gaming industry. It is a VERY relevant clinical tool!
l 74-75: That's where I am confused because you say for example that optoelectronic motion capture systems are low cost, time efficient etc... which you said the opposite just before.

l 75-77: Could you please add references cause I don't think I have seen inertial sensors been used in the clinic yet (or not for 3D gait analysis).
line 78: What do you mean by a segmental approach? Could you please elaborate?

l 78-81: I am not sure I follow what you are trying to say here. First are you talking about 3D gait analysis (cause you mention kinematics)? what do you mean by considers the movement of each body segment segment separately? We never look at 1 body segment, we usual are interested in joints and each joint is linked by 2 body segments. For example, the knee joint is liked by the thigh and the shank. To describe knee joint motion we usually look at the shank motion relative to the thigh.. Kinematics are not measured at the centre of mass. I am unsure what you are trying to say here.

l 84-85: Your sentence is very wrong. I am not familiar with harmonicity of cyclic movement (which you should try to explain better in the introduction) but IMUs are doing the job just fine to estimate spatiotemporal parameters as well as force plates, pressure mats and traditional 2D video cameras.

Materials and methods
I would like to have more anthropometric information on your participants.

In your inclusion criteria you mention that the participants should not have medical condition that might affect normal gait pattern and then you mention neurological conditions such as Alzheimer, parkinson and dementia. What about musculoskeletal conditions such as osteoarthritis, hip/knee replacement surgeries etc...?

I am not familiar with your optical sensors system Optogait. I think it would be good if you can detail a bit more on how it works and the output you gait cause I am not sure I understand fully. As you wrote it it looks like a motion capture system which it is not at all.

As for spatiotemporal parameters I am more familiar with step width, step length, stride length, cadence, stride time and velocity (which are the clinical parameters that most clinicians observe). Why you did not include these in your analysis and why concentrating on step and stride frequency? How does that relate to clinical spatiotemporal parameters and how can clinician interpret it?

60Hz seems a very low frequency for gait analysis. Understanding that in 1 second we can make about 2 steps which means that you will have about 30 data points per step (not even stride) which means you will miss some important information.

Results
Anthropometric measurements of the participants should not be in the result section but in the material and methods.

I don't understand why you did a statistical analysis looking at the difference between male and female. It is well known that female are in average shorter than males and therefore their stride length will be shorter as well. Moreover I don't even understand what is the scientific reason for doing such a statistical analysis, what was your research question for that analysis? Are male and female different? We already know that.

I don't understand what F1 and F2 represent scientifically and clinically.

Discussion
As per my comments before I do not believe that your software is identifying the main quantitative characteristics of normal gait. Stride and step frequency are far from being the main characteristics of gait. I am not sure how a clinician is supposed to interpret stride and step frequency and harmony, even less your so called F1 and F2 parameters.

Your video analysis software is extremely dependent on the lab/clinic lighting setting, how can you make sure that your detection will be as accurate in any setting?

I don't see your study limitations in your discussion section.

Author Response

The authors present a video-based software that can calculate some parameters, called F1 and F2, which are correlated to step and stride frequency, stride variability and gait harmony. I have a lot of issues in understanding their methodology and how F1 and F2 are clinically relevant for clinicians. I don't really understand the need for their software and the validation of their software output is questionable.

Below are the comments

We thank the reviewer for the constructive criticism. Based on the detailed comments, we have the impression that part of the difficulty in understanding our manuscript could be related to the differences in technical terminology in the engineering vs the medical field. In the point-by-point answers below and throughout the manuscript we have attempted to reconcile the apparent differences and, hopefully, improved the clarity of our manuscript to all readers.

Abstract: 
You should define stride frequency (SF) and step frequency (SPF) in the abstract.

You forgot to define CV

Thanks for spotting these omissions. As suggested, we defined stride frequency, step frequency/cadence and “CV” in the abstract.

Introduction:
l 42-45: This one sentence is a bit long and complicated and does not make much sense to me. Can you try to break it down.

Lines 42-45 were rephrased to improve clarity.

line 50: You should define stride frequency (SF) and step frequency (SPF)

We defined stride frequency and step frequency. In addition, to align with a more medical terminology, we clarified that step frequency is a synonym of cadence.

line 59: Missing reference after your statement on harmony associated with risk of falling in older adults.

Correct. We added at line 67 two references about the association between harmony and risk of falling in older adults:

  • Menz, H. B., Lord, S. R., & Fitzpatrick, R. C. (2003). Acceleration patterns of the head and pelvis when walking are associated with risk of falling in community-dwelling older people. The Journals of Gerontology Series A: Biological Sciences and Medical Sciences, 58(5), M446-M452.
  • Howcroft, J., Kofman, J., Lemaire, E. D., & McIlroy, W. E. (2016). Analysis of dual-task elderly gait in fallers and non-fallers using wearable sensors. Journal of biomechanics, 49(7), 992-1001.

line 61: I would not say that the term harmonicity is common in gait analysis

We accept the reviewer suggestion and modified the text accordingly. In addition, we preferred the term harmony as opposed to harmonicity.

l 66-68: I am really surprised to hear that stereophotogrammetry is a gold standard in gait analysis. I have been working in that field for some time and never ever heard of it for gait analysis. Moreover the authors cite 3 papers to ascertain their point but non of the 3 papers refer to stereophotogrammetry systems.

The reviewer remark allowed us to understand that there may be a definition/terminology issue across different field of research. Stereophotogrammetry is a term mostly used in the engineering field that refers to the specific type of optical motion capture system, working with infrared cameras that record the position in space of retroreflective (i.e. passive) markers. This is in fact the gold standard approach to gait analysis.

In the clinical field, the very same technique is more commonly referred to as “optical motion capture system”. This is in fact an umbrella term, that refers to a wider range of systems (comprising stereophotogrammetry) that provide 3D gait analysis through the detection of markers (active or passive) taped on specific body landmarks.

Having understood that the term stereophotogrammetry is a potential source of confusion, we decided to replace the term with “optical motion capture system”, that will be more recognisable by all readers.

l 66 - 74: I believe you are confusing 2D and 3D gait analysis. With 2D gait analysis you can easily extract spatiotemporal parameters (you don't even need cameras just a pressure mat will be enough). However to describe kinematics and kinetics (3D gait analysis) you will need an optical motion capture system with spherical markers taped to specific body landmarks (or active markers which is less common). That will be your 3D gait analysis gold standard to compute kinematics. Then you can add force plates to compute kinetics. I hope it makes more sense than your paragraph.

We are in full agreement with the definition of 2D and 3D gait analysis provided by the reviewer and there is no confusion about this. Again, we think there is a terminology problem. What we call “stereophotogrammetry” (specific term, engineer environment) is more commonly known as “optical motion capture system” (general term, clinical environment). We have changed the text substituting stereophotogrammetry with optical motion capture system, that will be more recognisable by all readers.

l 69: What do you mean high frequency? Usually the cameras will record at 200Hz (which I believe is not that high frequency, you can lower it down to 100Hz if needed). The force plates on the other hand needs really high frequency to capture heel strike (usually 1000Hz).

We understand that this adjective may have caused confusion. When we talk about frequency in video analysis we usually refer to frame rate that is the number of frame per second (fps). The standard rate in all modern movies and TV shows is 24 fps/Hz. This rate allows the human eye to perceive a fixed sequence of images as motion. 

However, for clinical or sport video analysis this frame rate is not enough to capture all that happens during a movement. For this purpose, high frequency cameras have been developed. The limit is given by the resolution of the image: usually higher frame rate means lower quality images namely images with less pixels. If our aim is for example to find when the heel strike happens, we must make a compromise between resolution and frequency.

In biomechanics most of the motion capture systems provide cameras with different frame rate that can assure different level of resolution: 240 fps/Hz with 8MP is usually a good compromise both in terms of video analysis and of cost of the equipment. 

As the reviewer correctly pointed out, when we talk about force plates, the frequency range is completely different as the system can reach up to 1000Hz (and some of the brands probably even more). In this case 200Hz would not be considered high frequency. The typical camera resolution requirement for motion capture analysis has been specified at lines 72-73, while removing the adjective “high” that had generated the confusion.

l 73-74: Most major clinics have an optical motion capture system and this system was initially developed for clinics and then was implemented in research facilities and for the gaming industry. It is a VERY relevant clinical tool!

We respectfully disagree with the reviewer on this remark. As useful as optical motion capture systems are, most major clinics do not have these systems and motion capture is not the standard of care worldwide. For example, it is not so in Europe in general and in Italy in particular. High costs of the equipment, time investment and complexity of experimental setup, data acquisition and processing prevent the application of this technology on a large scale. This is the very reason why more accessible alternatives, such as the approach proposed in our study, are sought.

Having said this, we have rephrased (lined 80-81) to take into account different contexts, included those in the reviewer’s experience.

l 74-75: That's where I am confused because you say for example that optoelectronic motion capture systems are low cost, time efficient etc... which you said the opposite just before.

Again, we realise that terminology is a source of confusion. We have now used the term “photoelectric sensors” throughout the manuscript.

l 75-77: Could you please add references cause I don't think I have seen inertial sensors been used in the clinic yet (or not for 3D gait analysis).

Thanks for the suggestion. Inertial sensors are indeed less common in clinical gait analysis but they are increasingly used for their time and cost-efficiency. As requested, we added the reference to a systematic review and meta-analysis: Petraglia, F., Scarcella, L., Pedrazzi, G., Brancato, L., Puers, R., & Costantino, C. (2019). Inertial sensors versus standard systems in gait analysis: a systematic review and meta-analysis. European Journal of Physical and Rehabilitation Medicine, 55(2), 265-280).

line 78: What do you mean by a segmental approach? Could you please elaborate?

The reviewer’s comment made us realise that we were not very clear about this point. Because human body is a complex multi-segment system and gait involves motion of the whole body, the biomechanics of multiple segments of the lower and upper body has been used to describe gait. Depending on which aspect of gait we want to focus on, we can measure biomechanical parameters of just one or more segments (e.g. cadence, stride length, knee/pelvis range of motion on the tree planes during distinct gait phases, acceleration patterns of the lower and upper trunk…). Furthermore, the study of the interaction, coordination and symmetry of the biomechanics of multiple body segments allows us to interrogate the harmony and quality of gait [Herran 2014] and the mechanisms of motor control [Williams 2017]. Traditionally, measures of coordination and symmetry are obtained through a segmental approach: the biomechanical interaction between two or three body segments, that are supposed to sufficiently represent the entire body, is typically explored (e.g. left/right symmetry and rhythmicity of spatiotemporal parameters, relation of harmonic index of accelerations of the pelvis and the lower and upper trunk). However, recent studies have proposed a more global approach to study the interaction between several body segments of the lower and upper body, to provide a global view of the whole body movements.

We have clarified the above concepts by modifying the corresponding paragraphs in the introduction. Hopefully these changes improved the clarity of the introduction.

l 78-81: I am not sure I follow what you are trying to say here. First are you talking about 3D gait analysis (cause you mention kinematics)? what do you mean by considers the movement of each body segment segment separately? We never look at 1 body segment, we usual are interested in joints and each joint is linked by 2 body segments. For example, the knee joint is liked by the thigh and the shank. To describe knee joint motion we usually look at the shank motion relative to the thigh.. Kinematics are not measured at the centre of mass. I am unsure what you are trying to say here.

Again, we are in agreement with the reviewer and the problem is in terminology. With the term “body segment” we refer to well-defined and localized areas of the body as joints or parts between joints (i.e. “body segments”). We rephrased in the text (lines 65, 75, 87, 295).

l 84-85: Your sentence is very wrong. I am not familiar with harmonicity of cyclic movement (which you should try to explain better in the introduction) but IMUs are doing the job just fine to estimate spatiotemporal parameters as well as force plates, pressure mats and traditional 2D video cameras.

We agree with the reviewer that inertial measurement unit sensors have been used to estimate spatiotemporal parameters of gait due to their low cost and easiness of used compared to the gold standard optical motion capture. However, the accuracy and precision of these systems for the study of spatiotemporal parameters in different forms of locomotion remains to be fully validated. In particular, not all IMUs are created equal, the acquisition frequency and calibration procedures playing a crucial role in the quality of the measurements [Petraglia et al., 2019].

Harmony is a different issue: there is no consensus on either the optimal definition of harmony or the reference method for its measurement. In this “work in progress” context, IMUs systems are increasingly applied thanks to their low cost and easiness. While this approach has not been specifically validated, it represents an example of a segmental approach to movement analysis. IMUs measure kinematics of the specific part of the body on which they are placed. The characterisation of within and between body segments coordination, left/right asymmetries and global quality of gait requires the simultaneous application of several sensors and is typically not performed.

We hope that these specifications will help the reviewer see our point of view and revise her/his initial judgement on our phrase.

Materials and methods
I would like to have more anthropometric information on your participants.

Participants had a mean(SD) weight of 69.7(13.0) kg and a height of 173.7(8.1) cm. These data were added to the manuscript.

In your inclusion criteria you mention that the participants should not have medical condition that might affect normal gait pattern and then you mention neurological conditions such as Alzheimer, parkinson and dementia. What about musculoskeletal conditions such as osteoarthritis, hip/knee replacement surgeries etc...?

The reviewer is correct. Subject’s evaluation upon enrolment included the identification of possible musculoskeletal conditions, current and past up to the last three years, that could affect gait or running pattern (e.g. osteoarthritis, hip/knee replacement surgeries). A positive current or past history of musculoskeletal injuries affecting gait was an exclusion criterion. We specified this in the text.

I am not familiar with your optical sensors system Optogait. I think it would be good if you can detail a bit more on how it works and the output you gait cause I am not sure I understand fully. As you wrote it it looks like a motion capture system which it is not at all.

As suggested by the reviewer, we added the following description in the Materials and Methods section. “Optogait is a system for the analysis of gait that is based on non-wearable floor sensors. Optical detection of foot position in a single plane is made through transmitting and receiving 1-meter bars placed on the sides of the treadmill tape, at the contact surface level, connected to a computer controlled by the researcher. Each bar contains 96 LEDs communicating on an infrared frequency with the same number of LEDs on the opposite bar. Once positioned on the treadmill, the system detects the interruptions of the communication between the bars (caused by the subject’s movement) and calculates the duration and position. During the execution of a gait, contact and stride times can be measured with an accuracy of 1 thousandth of a second and the position of the interrupted LEDs with a space resolution of 1,041 cm. Data were recorded continuously, at a frequency of 1000 Hz, through the dedicated software provided with the system which measures in real-time a series of crucial data for the movement analysis, such as step length, stride length, cadence, stride time and velocity.  Software analysis settings for the identification of gait events were optimized based on pilot testing with concurrent video recording choosing a 1_1 filter setting.”

As for spatiotemporal parameters I am more familiar with step width, step length, stride length, cadence, stride time and velocity (which are the clinical parameters that most clinicians observe). Why you did not include these in your analysis and why concentrating on step and stride frequency? How does that relate to clinical spatiotemporal parameters and how can clinician interpret it?

The reviewer raises an interesting point. We hope that it is now clear that step frequency is in fact a synonym of cadence. In the revised version of the manuscript we have substituted the step frequency with the more common term cadence.

Step and stride length are directly measured by the Optogait system but not by the new method. However, they can be easily calculated based on F1 and F2, as derived by our Markerless Technique for Quantitative and Qualitative Gait Analysis, as follows:

Step length (m)= speed (m/s) / F2 (Hz)

Stride length (m) = speed (m/s) / F1 (Hz)

The validity of step/stride lengths, as calculated based on the measured frequencies, can only be as good (or as bad) as the original measure. Therefore, in this shot communication (i.e. letter) we decided to limit the comparison between methods to the directly measured variables (i.e. stride and step vs F1 and F2 frequencies).

As for step width or contact time, these variables can’t be measured through our approach.

In conclusion, our new markerless video based technique measures some but not all the spatiotemporal indexes that can be used for gait analysis in clinic. However, the indexes provided (i.e. stride frequency, cadence, step/stride length, index of harmony) allow to perform an easy and cost-efficient first level diagnostic and possibly identify the indication for a second level evaluation.

60Hz seems a very low frequency for gait analysis. Understanding that in 1 second we can make about 2 steps which means that you will have about 30 data points per step (not even stride) which means you will miss some important information.

We respectfully disagree. 60 Hz is enough when we want to evaluate the frequency content of a normal gait pattern. 60 Hz means 60 samples per second and this allow us to appreciate difference up to 0.017 seconds. Even if wanted to extrapolate from a frame of the video the exact moment of contact of the foot to the ground, which is not our purpose, the error would be really small.

Furthermore, as the reviewer has correctly pointed out, the normal step frequency is around 2 Hz. According to the sampling theorem (Shannon, 1949), to reconstruct a one-dimensional signal from a set of samples, the sampling rate must be equal to or greater than twice the highest frequency in the signal. Even considering that someone can walk with a bigger frequency, 60 Hz is at least 10 times greater the frequency of interest.

Results
Anthropometric measurements of the participants should not be in the result section but in the material and methods.

As suggested, we moved anthropometric measurements from the results section to material and methods.

I don't understand why you did a statistical analysis looking at the difference between male and female. It is well known that female are in average shorter than males and therefore their stride length will be shorter as well. Moreover, I don't even understand what is the scientific reason for doing such a statistical analysis, what was your research question for that analysis? Are male and female different? We already know that.

Also based on the suggestion of reviewer # 1, we have realised that we may have excessively emphasised the male/female comparison, that is in fact not the focus of the study. Therefore, while we presented the spatiotemporal gait parameters separately for males and females to provide sex norms for healthy gait, we have decided to remove the between sexes comparison.

I don't understand what F1 and F2 represent scientifically and clinically.

From a practical stand point, our data demonstrate that, for normal gait, the dominant frequencies identified by Graal analysis coincide with stride frequency (F1) and step frequency or cadence (F2).

From a methodological point of view, F1 and F2 are the two main frequency contents derived from the spectral analysis of the variation of brightness over time of the pixels of the image of a person who walks. These are the most important (or “dominant”) frequencies that appear in a large proportion of pixels of the image. The higher is the percentage of pixels associated with a given frequency, the greater is the portion of the body image moving at unison at that frequency.

We hope this helps to clarify how the parameters are calculated and their meaning in practice.

Discussion
As per my comments before I do not believe that your software is identifying the main quantitative characteristics of normal gait. Stride and step frequency are far from being the main characteristics of gait. I am not sure how a clinician is supposed to interpret stride and step frequency and harmony, even less your so called F1 and F2 parameters.

As underlying before, our new markerless video based technique measures some (i.e. stride frequency, cadence, step/stride length), but not all the quantitative indexes that can be used for gait analysis in clinic. We hope that the previous paragraphs have clarified that F1 coincides with stride frequency and F2 with step frequency or cadence and that step and stride length can be calculated accordingly. Furthermore, our approach provides a promising index of gait quality, i.e. index of harmony.

From a practical stand point, time-efficiency and cost-efficiency are key aspects: all it takes is a 30s of video recording of treadmill walking, with a 60 Hz camera or phone. The video can be uploaded and analysed in about 10 min. We hope that the reviewer can agree that the possibility to derive this simple yet valid information with a 15-min time investment and a negligible cost is valuable for a first level diagnostic and/or longitudinal monitoring.

Your video analysis software is extremely dependent on the lab/clinic lighting setting, how can you make sure that your detection will be as accurate in any setting?

The reviewer poses an interesting point. Data collection for this study was conducted in two different settings and both ambient were well illuminated by standard artificial light with a top-down direction. Our method is based on the spectral analysis of the fluctuations of brightness over time of the pixels that compose the image of a person while walking. As such, the method should be relative independent from the absolute ambient light. Based on the reviewer’s comment we went back to the data and established that average pixel brightness from the two settings was significantly different, as shown in the box plot in the attachment: in setting #Promotus mean(SD) brightness was 110.53(32.37)RGB, in setting #DonGnocchi 126.25(46.81)RGB (p<0.0001).

However, the correspondence between stride/step frequency and F1/F2 was high and not different among settings. Correlation coefficients between stride/step frequency and F1/F2 for setting #Promotus were respectively 0.967 and 0.981, while for setting #DonGnocchi were 0.987 and 0.986. Based on the above, it would seem that the performance of the method is apparently robust and, at least in well illuminated ambient, unaffected by brightness level. However, the validity of this approach in a sub-optimally illuminated or else in an unusually bright ambient remain to be verified. We added this last consideration in the limitations section of the study (lines 322-326).

I don't see your study limitations in your discussion section.

As suggested by the reviewer, we added the study limitations at the end of the discussion section (lines 322-326).

Round 2

Reviewer 3 Report

Introduction
lines 42 to 45: Your sentence is still too long. Break it down into 1 sentence about the ability of walking. One sentence about quantitative gait analysis. I don't think that clinicians "craft" therapeutic interventions. Wrong verb here.
line 50: Again you miss on defining SPF.

Methodology:
Line 118: Subjects walked at an average self-selected speed ...
Line 119: You did not define nSL before using it. Also you used "," to separate decimals which is not the international convention "."
Line 120: Which gait parameters? Were analyzed using through the use (please rephrase your sentence).
Line 152: Do you mean it was synchronized with Optogait?
Line 220: You misspelled Shapiro-Wilk

Results:
Based on your answer to my comments you wrote that you presented the spatiotemporal gait parameters separately for males and females to provide sex norms. My problem is you only have 20 females, what kind of norm is that? You should not compare male and female in this study as it is out of context. You want to validate your web-based software not on a gender basis but representative of a healthy population!
You did not answer my question, what is the clinician suppose to do with F1 and F2, how can you interpret it clinically? I understand F1 and F2 are correlated (not strongly) with stride frequency and cadence but it does not give a clinically relevant index for clinician about what happens with their patient.
Are you familiar with the work from Altman and Bland? Their paper is very popular when comparing 2 clinical measurements or in your case comparing a new measurement technique with a gold standard. Altman DG, Bland JM (1983). "Measurement in medicine: the analysis of method comparison studies". The Statistician. 32 (3): 307–317
I highly recommend that you add Bland-Altmann plots for F1-SF and F2-CAD
Figure 3: Careful you use the term SPF in your graph. Also you use the "," sign to separate decimals but the convention is to use "." (same for figure 4)
Figure 4: I would not say that a r2 of 0.13 is moderately correlated but actually there is not much correlation (FYI: R2=0 means there is no correlation). It means that 13% of your harmony index can explain SV (which is not big).

Discussion
Line 266: HI is not correlated to SV!
Lines 272 to 276: Did you do a power analysis to back up your theory that 86 individuals is enough to provide reference values for the population, especially 20 females to represent all females in the population?
Line 296: Again I highly disagree with you a R2=0.13 means there is very little correlation. This is highly misleading! You should rewrite your entire paragraph cause this is scientifically untrue that SV and HI are correlated.
